# Genome-Wide Analysis of the Protein Phosphatase 2C Genes in Tomato

**DOI:** 10.3390/genes13040604

**Published:** 2022-03-28

**Authors:** Jianfang Qiu, Lei Ni, Xue Xia, Shihao Chen, Yan Zhang, Min Lang, Mengyu Li, Binman Liu, Yu Pan, Jinhua Li, Xingguo Zhang

**Affiliations:** 1Key Laboratory of Horticulture Science for Southern Mountainous Regions, The Ministry of Education, College of Horticulture and Landscape Architecture, Southwest University, No. 2 Tiansheng Road, Beibei, Chongqing 400715, China; qjf991314@163.com (J.Q.); nl1033148752@163.com (L.N.); xiaxue.tete@outlook.com (X.X.); chenshihao012@163.com (S.C.); zy778865@email.swu.edu.cn (Y.Z.); lang196518@email.swu.edu.cn (M.L.); mengyu151545@163.com (M.L.); swu_liubinman@126.com (B.L.); pany1020@swu.edu.cn (Y.P.); ljh502@swu.edu.cn (J.L.); 2State Cultivation Base of Crop Stress Biology for Southern Mountainous Land of Southwest University, Academy of Agricultural Sciences, Southwest University, Beibei, Chongqing 400715, China

**Keywords:** tomato, protein phosphatase 2C, genome-wide, *Ralstonia solanacearum*

## Abstract

The plant protein phosphatase 2C (PP2C) plays an irreplaceable role in phytohormone signaling, developmental processes, and manifold stresses. However, information about the *PP2C* gene family in tomato (*Solanum lycopersicum*) is relatively restricted. In this study, a genome-wide investigation of the *SlPP2C* gene family was performed. A total of 92 *SlPP2C* genes were identified, they were distributed on 11 chromosomes, and all the SlPP2C proteins have the type 2C phosphatase domains. Based on phylogenetic analysis of *PP2C* genes in *Arabidopsis,* rice, and tomato, *SlPP2C* genes were divided into eight groups, designated A–H, which is also supported by the analyses of gene structures and protein motifs. Gene duplication analysis revealed that the duplication of whole genome and chromosome segments was the main cause of *SLPP2Cs* expansion. A total of 26 cis-elements related to stress, hormones, and development were identified in the 3 kb upstream region of these *SlPP2C* genes. Expression profile analysis revealed that the *SlPP2C* genes display diverse expression patterns in various tomato tissues. Furthermore, we investigated the expression patterns of *SlPP2C* genes in response to *Ralstonia solanacearum* infection. RNA-seq and qRT-PCR data reveal that nine *SlPP2Cs* are correlated with *R. solanacearum*. The above evidence hinted that *SlPP2C* genes play multiple roles in tomato and may contribute to tomato resistance to bacterial wilt. This study obtained here will give an impetus to the understanding of the potential function of *SlPP2Cs* and lay a solid foundation for tomato breeding and transgenic resistance to plant pathogens.

## 1. Introduction

Plants may inevitably encounter many kinds of unpredictable environmental challenges, such as pathogen infection, extreme temperature, salt, and drought, which will adversely affect their growth, development, and production [1]. To adapt to these adversities, plants have evolved signaling mechanisms to deliver stimuli to different cellular compartments and then feedback to these stresses. Protein kinases (PKs) and protein phosphatases (PPs) modulate the protein function by reversible protein phosphorylation mechanism and are known to play a vital role in pivotal stress signaling processes [2]. In the past years, several PKs had been massively investigated and proved to be positive regulation factors responding to a variety of stresses [3,4,5,6]. On the contrary, PPs have not been studied as extensively as PKs.

PPs can remove phosphate groups from phosphorylated proteins through their special structure and can change protein function to respond to external pressure [7]. They are categorized into three major classes: tyrosine phosphatases (PTPs), serine/threonine phosphatases (PSPs), and dual-specificity phosphatases (DSPTPs), which is determined by the specificity of a substrate [8]. Moreover, PSPs are further classified into phosphoprotein phosphatases (PPP) and phosphoprotein metallophosphatase (PPM) based on distinct amino acid sequences, different dependencies on metal ions, and sensitivities to inhibitors [9]. PP1, PP2A, PP2B, PP4, PP5, PP6, and PP7 were included in the PPP family, whereas the PPM family is represented by PP2C and others [8,10,11].

Due to the special differences between amino acid and crystal structure, PP2C proteins require metal ion Mg^2+^ or Mn^2+^ to accomplish their function [12]. In eukaryotes, the catalytic domain of PP2C proteins is located at either the N-terminus or the C-terminus. Further research revealed that the regions of the catalytic domain in eukaryotic PP2C proteins are relatively conserved, whereas the regions of a non-catalytic domain have diverse amino acid sequences [13]. PP2Cs are quite conserved in evolution from prokaryotes to higher eukaryotes, having been found in archaea, bacteria, fungi, plants, and animals, significantly modulate stress signaling pathways, and reverse the stress-induced PK cascades in response to environmental stimuli [14]. In plants, *PP2Cs* form the largest family of phosphatase genes, accounting for 60–65% of all phosphorylases [8,15]. The high proportion of *PP2C* genes in plants indicates their evolutionary significance, requirement, and participation in varying plant cellular functions [10]. As a major class of protein phosphatases, PP2Cs catalyze the dephosphorylation of substrate proteins to regulate signaling pathways and participate in various physiological and biochemical processes in plants. Current studies have indicated that PP2Cs play important regulatory roles in different processes, such as ABA signaling, biotic and abiotic stress responses, plant immunity, K^+^ nutrient signaling, and plant development [8,10,13,14,15,16,17]. There were extensive studies on the role of PP2C proteins. For example, *PpABI1A* and *PpABI1B* in moss are directly involved in ABA responses [18]. In higher plants, the function of *PP2Cs* is more diverse, among which *PP2Cs* in group A is the most intensively studied. In *Arabidopsis thaliana*, the PP2Cs of group A, well-studied as ABA co-receptors, negatively regulate the ABA signaling pathway. For instance, *ABI1*, *ABI2,* and *HAB1* participate in plant abiotic stress by negatively regulating ABA signaling and *ABI1* plays a negative regulatory role in response to ABA-mediated drought stress [19,20,21]. In tomato, *SlPP2C1*, an *ABI2* homology, negatively regulates ABA signaling and fruit ripening. *SlPP2C1-RNAi* led to increased endogenous ABA accumulation and advanced release of ethylene in transgenic fruit compared with wild-type (WT) fruit. *SlPP2C1-RNAi* plants were hypersensitized to ABA and displayed delayed seed germination and primary root growth, and increased resistance to drought stress compared with WT control plants [22]. In maize, *ZmPP2C-A10* is tightly associated with drought tolerance, and similar to its Arabidopsis orthologs, it interacts with *ZmPYL* ABA receptors and ZmSnRK2 protein kinases, suggesting that *ZmPP2C-A10* is involved in mediating ABA signaling. Transgenic studies confirmed that *ZmPP2C-A10* functions as a negative regulator of drought tolerance [23]. Similar results have been obtained from poplar [24] and sweet cherry [25] studies. On the contrary, in *Brachypodium distachyon*, *BdPP2CA6,* a member of group A of PP2C, was found to be a positive regulator in both ABA and stress signaling pathways [26]. These studies indicate that the PP2Cs of group A have diverse functions in different plants. In Arabidopsis, the PP2Cs of group B function as mitogen-activated protein kinase (MAPK) phosphatases. *AtAP2C1* regulates phytohormone and defense responses by cooperating with MPK4 and MPK6 [27,28,29], whereas *AtAP2C3* mediates stomata development; thus, negatively regulating MAPK signaling [30]. Group D comprises nine *PP2Cs* in Arabidopsis, all of which have different expression patterns and subcellular localization [31]. In group E, *AtPP2C-6-6* interacts with histone acetyltransferase (*AtGCN5*) to control the activation of stress-responsive genes in the stomatal signaling network. In group F, *AtWIN2* may interact with the bacterial effector *HopW1-1* and regulates *HopW1-1* to induce disease response [32]. These studies indicate that *PP2Cs* have diverse functions worth investigating.

To date, there are 80, 78, 257, 62, and 131 genes coding for PP2C proteins identified using bioinformatics surveys in Arabidopsis, rice [33], wheat [34], woodland strawberry [35], and *Brassica rapa* [36], respectively. The studies mentioned above have proved the diverse roles of *PP2C* genes. Hence, it is necessary to delve into the comprehension and functional characterization of the *PP2C* gene family.

Tomato is one of the most consumed vegetables in the world, and its annual output has reached 181 million tons (Food and Agriculture Organization of the United Nations, http://www.fao.org/ (accessed on 5 March 2022)). Tomato and its derivatives have become an indispensable part of daily life. *Ralstonia solanacearum*, previously known as *Pseudomonas solanacearum*, is internationally recognized as one of the leading models in the analysis of plant pathogenicity. This soil bacterium is a severe and devastating disease of solanaceous crops (https://iant.toulouse.inra.fr/R.solanacearum (accessed on 5 March 2022)). Tomato genome sequencing has been completed (https://solgenomics.net (accessed on 5 March 2022)), but the tomato PP2C family genetic research in plant pathogen, *R. solanacearum*, remains unknown. In this study, bioinformatics analysis of tomato PP2C gene family members was conducted to preliminarily explore the expression characteristics and response rules of this gene family in tomato which were infected with *R. solanacearum*. In our study, 92 *SlPP2Cs* were identified, and their physical and chemical properties, subcellular localization, exon–intron structure, phylogenetic comparison, gene duplication, chromosome distribution, and cis-acting elements were analyzed. *SlPP2Cs* under *R. solanacearum* infection were also examined. Our results were a reliable prediction for the function and structure of *SlPP2Cs*, which would provide a solid basis for performing further functional analysis of these genes.

## 2. Materials and Methods

### 2.1. Identification of SlPP2C Members

To gain the whole *SlPP2C* genes of tomato, genome data of ITAG 4.0 were downloaded from Phytozome (http://phytozome.jgi.doe.gov/pz/portal.html (accessed on 5 March 2022)) [37] to set up a local database, and seed file of the SlPP2C protein domain (PF00481) [38] was downloaded from the Pfam (http://pfam.xfam.org/ (accessed on 5 March 2022)) [39]. The HMMER [40] procedures of hmm build and hmm search were fully used to retrieve all assumed *SlPP2C* sequences with default parameters in tomato and the ID of the relevant sequences was collected (E-value < 1.0). The ID was submitted to the SGN (http://solgenomics.net/ (accessed on 5 March 2022)) [41] to eliminate the sequence that shows no difference in their amino acid. After manually removing the redundant sequences, all these predicted genes were examined in SMART (http://smart.embl-heidelberg.de/ (accessed on 5 March 2022)) [42] and CDD (https://www.ncbi.nlm.nih.gov/cdd (accessed on 5 March 2022)) [43]. Protein physicochemical properties and subcellular localization of *SlPP2Cs* were calculated by the online software ExPASy (https://www.expasy.org/ (accessed on 5 March 2022)) [44] and Cell-PLoc 2 (http://www.csbio.sjtu.edu.cn/bioinf/plant-multi/ (accessed on 5 March 2022)) [45]. The chromosome length of tomato and the location data of each *SlPP2C* gene were retrieved from SGN.

### 2.2. Phylogenetic Analysis

Validated tomato PP2C protein sequences were employed for establishing an evolutionary relationship with the known PP2C members of Arabidopsis (*AtPP2Cs*) and *Oryza sativa* (*OsPP2Cs*) [33]. This analysis included 250 amino acid sequences. ClustalW [46] was used to conduct multiple sequence alignment. Ground on the outcomes of aa sequence alignment, the maximum likelihood method in MEGA-X [47] was used to design the phylogenetic tree with default parameters. The Blast program was applied to discern tandem repeat genes. If the identity between two genes was more than 75% and the alignment length was more than 75% of the longer sequence, they were considered to be tandem repeat gene pairs [48]. *Ka/Ks* of all *SlPP2C* tandem repeat gene pairs was calculated using KaKs_Calculator [49]. The relationship of tandem repeats was displayed via TBtools [50].

### 2.3. Chromosomal Location, Gene Structure, and Sequence Alignment

The *PP2C* genes were mapped to tomato chromosomes by identifying their chromosomal positions according to the SGN database. The proportion chart of the chromosomal location was drawn using TBtools. The exon and intron structures of *SlPP2Cs* were generated using the Gene Structure Display Server 2.0 (http://gsds.cbi.pku.edu.cn (accessed on 5 March 2022)) [51] by aligning the CDS sequences with the corresponding genomic DNA sequences from the SGN database. Domains were confirmed using the Pfam and the SMART programs.

### 2.4. Cis-Element Prediction for PP2C Gene Promoter

It is well known that most of the functional cis-active elements in vivo (about 80%) are concentrated in the proximal promoter [52]. However, how to determine the boundary of the promoter region has not been determined. Generally speaking, 1 kb, 2 kb, or longer upstream of ATG is taken. Therefore, the aim of this paper was to take 3 kb. The promoter sequence (3 kb upstream of the putative genes ATG) was extracted from the ITAG 4.0 gff3 file. All the promoter sequences were uploaded to the PlantCARE database (http://bioinformatics.psb.ugent.be/webtools/plantcare/html/ (accessed on 5 March 2022)) [53] for cis-element prediction. The result was visualized with TBtools.

### 2.5. Expression Analysis of SlPP2Cs of Tomato Tissue

RNA-seq data from the platform TFGD (Tomato Functional Genomics Database) (http://ted.bti.cornell.edu/ (accessed on 5 March 2022)) [54] played a part in the expression patterns of *SlPP2Cs*. The data included reads from the Illumina RNA-seq analysis of leaves, roots, flower buds, fully opened flowers, and 1, 2, and 3 cm, mature green, breaker, and breaker +10 fruit of the tomato cultivar Heinz. Gene expression level was defined on the basis of the normalized expression value, that is, reads per kilobases per million (RPKM) for each tissue/stage. The *log2* logarithmic transformation of the RPKM values was selected from the platform, and heat maps were plotted to analyze their expression levels.

### 2.6. Transcriptional Profiling of SlPP2C Genes in Tomato Infected with Ralstonia solanacearum

RNA-seq data from our laboratory (unpublished data) were performed to gain insight into the expression profiles of the *SlPP2C* gene families under *R. solanacearum* infection. Gene expression level was defined on the basis of the normalized expression value, that is, reads per kilobases per million (RPKM) for each sample. The *log2* logarithmic transformation of the RPKM values was selected, and heat maps were plotted to analyze their expression levels. The expression values of *SlPP2C* genes that were upregulated or downregulated by more than two-fold with *p* < 0.05 were considered as differentially expressed.

### 2.7. Bacterial Strain of Ralstonia solanacearum

The standard strain GMI1000 of *R. solanacearum* was used to infect tomato stems. The tested pathogen GMI1000 was grown on solid TTC medium plates for 3 days at 28 °C; then inoculated in liquid TTC medium to grow overnight at 28 °C [55]. Then, 100 μL overnight bacterial solution was absorbed into TTC liquid medium and activated for 24 h at 28 °C, which can be used for the inoculation test.

### 2.8. Plant Material and Treatments

The germinated seeds of tomato were grown in plastic pots containing a mixture of soil and vermiculite (3:1). The pots were then placed in greenhouse with a 16 h light/8 h dark cycle photoperiod, and the temperature was 25 ± 2 °C. The humidity was maintained at approximately 60–70%, and the photosynthetic photon flux density was controlled at about 120 μmol photons/m^2^/s. When the seedlings were six weeks old, the plants were used in the inoculation experiment. Untreated plants were used as controls to avoid the effects of biological clock on differential gene expression. Specific treatments were provided to the seedlings as follows: The stem of selected tomato plants was needled with 1 mL of activated bacterial solution and cultured in an artificial climate chamber with 28 °C for 3 days. After 3 days of treatment, the materials of stems were immediately frozen in liquid nitrogen and then stored at −80 °C until use.

### 2.9. RNA Isolation and Real-Time PCR

A total of 100 uL of RNA was extracted using the Total RNA Kit (BioTeke Corporation, Beijing, China), following the manufacturer’s instructions. The integrity of the RNA was verified by agarose gel electrophoresis. Synthesis of the cDNA was performed from the total RNA samples using the PrimeScript™ RT Reagent Kit, according to the protocol with gDNA Eraser (TaKaRa, Dalian, China). Specific primers were designed using qPrimerDB [56] and are presented in Appendix A. The *SlELF-α* gene was used as the internal control [57] to quantitate the expression of *SlPP2C* genes. Real-time PCR was performed using CFX96 Touch™ Real-time PCR System (Bio-Rad, Hercules, CA, USA) with a SYBR Premix Ex Taq™ II Kit (Bio-Rad). The reactions were carried out in the following conditions: denaturation at 94 °C for 4 min, 40 cycles of 5 s at 95 °C, 30 s at 60 °C, 15 s at 95 °C, 20 s at 60 °C, and 15 s at 95 °C. Three biological duplications were used. The 2^−∆∆Ct^ method was used to analyze the real-time PCR data [58]. Relative expressions were visualized using Graphpad Prism [59].

## 3. Results

### 3.1. Identification of PP2C Genes in Tomato

To identify the *PP2C* genes in the tomato, we searched for sequences that contained the particular domain in the tomato protein database using the hidden Markov model (HMM) model of PF00481 and confirmed the presence of PP2C domains using Pfam and Batch CD-search and found 92 *PP2C* genes (Appendix A). These genes were labeled as *SlPP2C01* to *SlPP2C92* on the basis of their distributions and relative linear orders among their respective chromosomes. The information of gene ID, the amino acid (aa) length, isoelectric point (pI), molecular weight (Mw), hydrophilic coefficient, instability index, and subcellular localization prediction of 92 PP2C proteins were analyzed (Table 1). The lengths of proteins varied from 59 aa residues (*SlPP2C62*) to 1080 aa residues (*SlPP2C37*), with an average length of 190 aa. Most of the lengths of the PP2C proteins were between 300 and 400 aa. The pI varied from 4.62 (*SlPP2C90*) to 11.39 (*SlPP2C08*) and Mw ranged from 6738.81Da (*SlPP2C62*) to 119917.55Da (*SlPP2C37*). The result of hydropathicity (GRAVY) showed that *SlPP2Cs* except for *SlPP2C92* were hydrophilic proteins. The result of the instability index proclaimed that 67.4% *SlPP2Cs* were unstable proteins. The result of subcellular localization prediction showed that most tomato PP2C proteins were predicted to be in the intracellular (97.8%), such as the cytoplasm, chloroplast, mitochondrion, nucleus, and peroxisome, but some proteins may be located in extracellular or cell membrane (2.1%). These results revealed that SlPP2C proteins were organelle-specific and had a function in various environments.

### 3.2. Phylogenetic and Comparative Synteny Analysis

To gain insights into the evolutionary relationship within the *PP2C* gene family in Arabidopsis, rice, and tomato, an unrooted phylogenetic tree was structured with the amino acid of 78 *OsPP2Cs* of rice, 80 *AtPP2Cs* of Arabidopsis, and 92 *SlPP2Cs* of tomato using the MEGA-X by the adopting maximum likelihood (ML) approach (Appendix A). The findings of the phylogenetic tree depicted that *PP2C* genes were further divided into eight groups, labeled from A to H, which was also supported by the analyses of *SlPP2Cs* gene structures and protein motifs. Group F is the largest, with 49 members, while group E is the smallest, with only 14 members (Figure 1, Table 2). The *SlPP2Cs* showed a species-specific evolutionary classification.

To better understand the difference in evolution and replication events involved in the *PP2C* gene family, the collinear relationship was analyzed in *PP2C* genes from tomato, rice, and Arabidopsis. The result showed that a total of 63 *PP2C* members participated in the synteny relationship (Figure 2, Appendix A). There are 17, 9, and 3 gene pairs represented as collinear in tomato, Arabidopsis, and rice, respectively, and the number of collinear gene pairs between tomato and rice, tomato and Arabidopsis, rice and Arabidopsis were 6, 12, and 5 pairs separately, which showed that *PP2C* genes of tomato, rice, and Arabidopsis reflected more diversity in evolution, and the number of homologous genes was relatively less (Appendix A).

Duplication genes and their synonymous (Ks) and nonsynonymous (Ka) substitution rates (*Ka/Ks**)* could reveal the evolutionary relationship and show the kind of selection pressure being encountered. *Ka*, *Ks* mutations, and *Ka/Ks* of 17 syntenic gene pairs of *SlPP2C* genes were calculated (Table 3, Appendix A). Commonly, if the value of *Ka/Ks* < 1, the duplicated gene pairs may evolve from purifying selection (also called negative selection); *Ka/Ks* = 1 means neutral selection; while *Ka/Ks* > 1 means positive selection [60]. The result proposed that 17 pairs of *SlPP2Cs* duplicated genes represented less than 1.00, suggesting that all duplicated *SlPP2C* genes have evolved mainly from purifying selection. We also calculated the divergence time (as T = *Ks/2λ*) among 17 pairs of duplicated *SlPP2C* genes based on a clock-like rate of 1.5 × 10^−8^ mutations per synonymous site per year, as proposed previously [61]. The result in Table 2 showed that divergence events of duplicated *SlPP2C* genes were estimated to have occurred around 2.33–73.33 Mya (million years ago). The average divergence time among these genes is 31.59 MYA.

### 3.3. Chromosomal Localization and Duplication of SlPP2C Genes

The 92 *SlPP2C* genes were mapped with the published chromosomes of the tomato genome to identify their distribution (Figure 3). They are scattered on 11 of the 12 chromosomes. Macroscopically, these *SlPP2Cs* were unevenly distributed across these chromosomes and mostly existed in the form of gene clusters. High-density regions harboring *PP2Cs* were found in chromosomes 01, 03, 06, 08, and 10 and discretely distributed in chromosomes 02, 04, 05, 07, 09, and 12. The most and the least *SlPP2Cs* were distributed on chromosome 03 (14 *SlPP2C* genes, accounting for 15.22%) and chromosome 02 (3 *SlPP2C* genes: 3.26%).

Previous studies in rice, Arabidopsis, and *B. distachyon* showed that *PP2C* gene families mainly expanded by whole-genome and chromosomal segment duplication [14,33]. Closely related genes located within a distance of fewer than 200 kb on the same chromosome are defined as tandem duplications; otherwise, they are segmental duplication [62]. According to this principle, 17 pairs of duplication *SlPP2C* genes were found to be involved in segmental duplication events and three of them may be involved in tandem duplication. These 17 pairs of duplicated *SlPP2C* genes are distributed on chromosomes 01, 02, 03, 04, 05, 06, 07, 08, 09, and 10, but not on chromosome 12. These three pairs of tandem duplication are distributed on chromosomes 03, 08, and 10.

### 3.4. Conserved Motif and Gene Structure of SlPP2Cs

In order to better understand the conservation and diversity of motif compositions and gene structures of *SlPP2Cs*, the conserved motifs and exon–intron organization of *SlPP2Cs* were analyzed and a new phylogenetic tree as a reference was also structured (Figure 4). The conserved motifs of SlPP2C proteins were analyzed using the software MEME and 10 distinct conserved motifs were identified (Table 4). In Figure 4A, the number of motifs ranged from 1 to 10 with 15–50 residues in all SlPP2C proteins. Motifs 1, 2, 3, 4, 5, 6, 8, 9, and 10 ubiquitously existed in SlPP2Cs, which showed that these motifs may have similar conserved positions and functions. Interestingly, motif 7 was selectively presented in a few SlPP2Cs, and they all belong to group H. Therefore, motif 7 may have an unusually special role in the process of regulation.

The whole *S**lPP2Cs* gene structures were analyzed using TBtools. As shown in Figure 4B, tomato *PP2C* gene exon–intron organizations were diverse. The difference in the number of exons (1–20) was apparent for *SlPP2Cs*. *SlPP2C74* was encoded by at least 20 exons which had the largest number of exons, whereas *SlPP2C51* only was encoded by one exon. Among the aligned 92 *SlPP2C* members, 98.91% of them had at least two introns except *SlPP2C51* where no introns were found. Furthermore, the members of the same group showed structural similarities, such as intron phase, intron number, and exon length. The result indicated that *SlPP2Cs* had evolutionary stability and versatility in tomato.

### 3.5. Cis-Elements in the Promoters of Tomato PP2C Genes

Gene transcription levels were regulated by the interaction of transcription factors with the cis-acting element in the upstream promoter sequences. Therefore, studying the cis-element in the promoter of *PP2C* genes in the tomato may help decrypt the function of *SlPP2C* genes. The promoter regions (3 kb upstream ATG) of 92 *SlPP2C* genes were analyzed using the online software PlantCARE. After excluding the common cis-elements, such as the TATA-box and CAAT-box, the remaining 26 cis-elements can be divided into six parts (Table 5). Then depicted by TBtools (Figure 5, Appendix A). Eight cis-elements, namely, GT1-motif, G-box, MRE, ACE, 3-AF1 binding site, Sp1, 4cl-CMA2b, and AAAC-motif are associated with light responsiveness. Ten cis-elements, including CGTCA-motif, TGACG-motif, ABRE, TGA-, AuxRR-core, TCA-, SARE, GARE-motif, P-box, and TATC-box are related to hormone induction. Two stress-related elements, including MBS and LTR, are induced by abiotic and biotic stresses. O2-site and MBSI have a connection with growth and development. Moreover, an element called Box III, which is a protein binding site, a regulatory element called MSA-like, and an element circadian-related circadian is found. Clearly, many cis-elements related to abiotic stress in plants were identified at the promoter of *PP2C* gene, and the promoter of 70 out of 92 *PP2C* genes had ABRE (cis-element involved in the ABA responsiveness), indicating that *PP2Cs* played a decisive role in abiotic stress resistance via ABA response.

### 3.6. Expression of SlPP2C Genes in Different Tomato Tissues

The processes that *SlPP2C* genes may be involved in tomato growth and development were studied. The RNA-seq data of 10 tissues/stages, including roots, leaves, flowers, flower buds, 1, 2, and 3 cm fruits, mature green fruits, breaker fruits, and fruits on day 10 after breaking in Heinz were downloaded from TFGD to analyze the tissue expression pattern of *SlPP2Cs* (Appendix A). As shown in Figure 6, *SlPP2Cs* have different expression levels in different tissues and stages, 17 genes, including *SlPP2C04*, *SlPP2C10*, *SlPP2C11*, *SlPP2C13*, *SlPP2C22*, *SlPP2C30*, *SlPP2C31*, *SlPP2C32*, *SlPP2C39*, *SlPP2C55*, *SlPP2C58*, *SlPP2C60*, *SlPP2C78*, *SlPP2C79*, *SlPP2C80*, *SlPP2C90,* and *SlPP2C91* were highly expressed in all tissues/stages. By contrast, 14 genes, including *SlPP2C03*, *SlPP2C07*, *SlPP2C08*, *SlPP2C12*, *SlPP2C17*, *SlPP2C18*, *SlPP2C19*, *SlPP2C23*, *SlPP2C24*, *SlPP2C26*, *SlPP2C62*, *SlPP2C68*, *SlPP2C69,* and *SlPP2C71*, showed low expression levels in all tissues/stages. The expression patterns of the other *SlPP2Cs* showed different patterns of temporal and tissue-specific expressions. *SlPP2C67* was highly expressed in roots and *SlPP2C91* accumulated in the buds. *SlPP2C27* and *SlPP2C39* were expressed in leaves. In addition, four genes named *SlPP2C09*, *SlPP2C13*, *SlPP2C30*, and *SlPP2C91* were abundant in fully opened flowers. *SlPP2C58* was highly expressed in break fruit and +10 break fruit. Intriguingly, these genes indicated a possible role in the organ development of tomato.

### 3.7. Transcriptional Profiling of SlPP2C Genes of Tomato Infected with Ralstonia solanacearum

*R. solanacearum*, originally named *Pseudomonas solanacearum*, is a destructive soil-borne plant pathogen [63]. With strong environmental adaptability and a wide host range, it can cause lethal wilting diseases of 200 plant species [63], ranking second among the 10 most harmful plant pathogenic bacteria worldwide [64]. Many important economic crops in China such as peanut, potato, tomato, tobacco, and banana are deeply affected by this pathogen [63]. Therefore, we sequenced the stem of tomato infected with *R. solanacearum*. RNA-seq data were drawn into a heat map to further examine the expression patterns of *SlPP2C* genes under *R. solanacearum* infection treatment. In Figure 7, under *R. solanacearum*-infection condition, a total of 65 *SlPP2C* genes (70.65%) were found (Appendix A), nine *SlPP2C* genes (*SlPP2C28*, *SlPP2C38*, *SlPP2C40*, *SlPP2C43*, *SlPP2C44*, *SlPP2C48*, *SlPP2C50*, *SlPP2C89*, *SlPP2C92*) were obviously upregulated and three *SlPP2C* genes (*SlPP2C14*, *SlPP2C27,* and *SlPP2C36*) were obviously downregulated. This indicated that the function of *SlPP2Cs* might be related to *R. solanacearum* resistance.

### 3.8. Analysis of SlPP2C Gene Expression in Tomato under Ralstonia solanacearum Infection

To further analyze the function of these gene families, qRT-PCR was used to investigate the expression of *PP2C* genes in tomato under *R. solanacearum* treatment. Based on the transcriptome data available in our laboratory, six upregulated and three downregulated expression genes were chosen. As shown in Figure 8, under *R. solanacearum* treatment, the expression levels of all five *PP2C* genes (*SlPP2C28*, *SlPP2C38*, *SlPP2C43*, *SlPP2C48*, and *SlPP2C92*) have varying degrees of increasing, the expression of *SlPP2C40* change is not significant, but that of all three *PP2C* genes (*SlPP2C14*, *SlPP2C27*, and *SlPP2C36*) significantly decreased. This is mostly consistent with our transcriptome data. The small difference may be due to the inconsistency between the RNA-seq and qRT-PCR samples. The above transcriptome and qRT-PCR data indicated that the *PP2C* gene may play an important role in resistance to plant pathogen infection. Therefore, these genes are worthy of further functional verification experiments.

## 4. Discussion

The *PP2C* gene family is one of the most significant gene families that plays a vital role in response to stresses such as drought, salt, alkali, fungal pathogens as well as in plant development [65]. Previously, many studies have been carried out on the functional analysis of *PP2C* genes in Arabidopsis [66] and tobacco [67]. To date, many *PP2Cs* have already been identified in maize [7], rice [33], wheat [34], Arabidopsis [33], hot pepper [68], woodland strawberry [35], and *Medicago truncatula* [69] using advanced techniques of bioinformatics. In this study, a comprehensive genome-wide analysis of the *SlPP2Cs* gene family was performed, including gene identification, phylogenetic relationships, evolutionary analysis, synteny relationships, chromosomal localizations, gene structures, conserved domains, and motifs. In addition, gene expression patterns of some key SlPP2C genes were also determined under pathogen stress conditions. Herein, a total of 92 SlPP2C genes were identified.

### 4.1. Evolution of the SlPP2C Gene Family

PP2Cs have been evolutionarily conserved from prokaryotes to higher eukaryotes. Compared to other gene families, the *PP2C* gene family is one of the largest families in the plant kingdom. In lower plants, such as *Chlamydomonas reinhardtii*, *Physcomitrella patens,* and *Selaginella tamariscina*, the PP2C gene family members are much less common than those in higher plants. The increase in the diversity and the total number of *PP2C* genes from lower plants to higher plants may correlate with adaptations to the environmental stresses [70].

Gene duplication is one of the main driving forces of biological evolution [71], and it may contribute to the diversity of *SlPP2C*. The prediction of evolutionary patterns by calculating *Ka/Ks* provides information about the type of selection, such as purifying, positive, and neutral selection of gene pairs during divergence [72]. In this study, 17 *SlPP2C* tandem repeat gene pairs were identified. *Ka/Ks* of these tandem repeats were calculated. The result showed that *Ka/Ks* for 17 pairs of duplicated *SlPP2C* genes was <1, suggesting that all duplicated *SlPP2C* genes have evolved mainly from purifying selection. This conclusion was mutually corroborated with the fact that the members of the *SlPP2C* gene family were conserved. In evolution, most of the genes copied from *SlPP2Cs* are adjacent to parental genes. Furthermore, for divergence time, we further utilized *Ks* values. The divergence time ranged from 0.07 to 2.2 (*Ks* values) and the mean duplication time is 31.59 MYA of these paralogous genes, which suggested that their divergence occurred later than the divergence time of Arabidopsis (about 16.1 MYA) [35]. The findings of our study demonstrated that Arabidopsis *AtPP2Cs* duplication time is much earlier than that of tomato *SlPP2C*. Therefore, the functional study of *AtPP2Cs* in Arabidopsis can provide a research basis for the study of *SlPP2C*.

### 4.2. Expression of SlPP2Cs

Environmental conditions cannot always maintain the optimal state needed for plant growth without artificial control. Therefore, plants are constantly challenged by a variety of environmental abiotic stress factors, such as drought, salt, high/low temperature, and biological stresses. These stresses seriously affect the yield and quality of tomato [73,74]. Until now, molecular mechanisms of plant responses to the above stress have been extensively studied. Plant hormones, such as ABA, SA, and GA, play a vital role in the ability of plants to cope with abiotic stresses by mediating growth, development, nutrient allocation, and source/sink transitions [75]. Among these, the significance of ABA signaling is well-documented in stress-adaptive modifications and stress resistance mechanisms. All *PP2C* group A genes in rice could be induced by ABA, and their relative expression levels were increased under high salt and low temperature treatment [33]. In Arabidopsis *ABI1*, *ABI2*, *HAB1*, *HAB2*, *AHG1*, and PP2C-A have reported encoding in ABA signaling networks [76,77,78,79,80]. Previous studies reported that PP2C regulates positively against salt tolerance in Arabidopsis and drought in peach to modulate the stress severity [81,82]. For example, the Arabidopsis *AtPP2CG1* positively modulates the abiotic stresses, including salt, drought, and ABA [81]. The *ZmPP2C-A10* gene has a negative regulatory role in maize response to drought stress [23]. Moreover, not all *PP2C* genes have a similar response to abiotic stresses. In Arabidopsis, two members of *PP2C* genes responded differently, such as *AP2C1* expression was strongly induced by cold, drought, and wounding, but *AP2C2* was slightly influenced under the same treatments, suggesting their functional diversity [83]. Arabidopsis *PP2C-D* is mainly expressed in the roots, while the wild soybean *PP2C-D* is mainly expressed in the stem. Therefore, the wild soybean *PP2C-D* and Arabidopsis *PP2C-D* have different regulatory roles in different stress. Some *PP2C-D* in wild and cultivated soybean are involved in different signal transduction pathways; thus, adapting to different resistance mechanisms [84]. These findings highlighted the significance of the *PP2C* gene family. Aiming to obtain gene expression patterns of *SlPP2C*, we downloaded the previously reported RNA-sequence data and analyzed the expression profile of *PP2C* genes. By comparing *PP2C* gene transcription profiling in 10 tissues/stages, including leaves, roots, flowers, flower buds, 1, 2, and 3 cm fruits, mature green fruits, breaker fruits, and fruits on day 10 after breaking, the expression of all the *SlPP2C* genes showed diverse tissue-specific patterns, such as some of the genes were highly expressed in all the tissues, while some expressed only in one and/or two tissues, intimating that *SlPP2C* proteins may play multiple roles in plants. Thus, their functions are worthy of further study.

The yield of crops is largely affected by different types of biotic stresses [85]. Many abiotic stress conditions have been shown to weaken the defense mechanism of plants and enhance their sensitivity to pathogen infection [86,87]. Thus, finding resistance genes that can resist pathogen infection is very important to improve crop yield. Previous studies have shown that PP2C proteins play multiple roles in plants. In the case of tomato infected with *R. solanacearum*, RNA-seq and expression pattern of some *SlPP2C*s were explored. *R. solanacearum* is a devastating soil-borne plant pathogen that brings serious losses to tomato production [63]. In our study, nine *SlPP2C* genes were upregulated under *R. solanacearum* infection treatment, showing that the *SlPP2C* gene family was indeed closely related to *R. solanacearum*. However, understanding whether or not each gene plays a critical role in abiotic stress tolerance still requires functional characterization of individual genes.

### 4.3. Possible Function of SlPP2Cs

The combination of our research and previous studies revealed that *SlPP2Cs* might have a variety of functions. The subcellular localization indicated that *SlPP2Cs* mainly had enzyme catalytic function in the cytoplasm and a few of them appeared in other cellular compartments, which may take part in other biochemical processes. For example, *AtAPD7*, an Arabidopsis *PP2C* protein, widely acted in the nucleus and cytoplasm of root cells and cytoplasm of mesophyll protoplasts [88]. *OsSIPP2C1* was located in the nucleus and it was negatively regulated by *ABL1* which could respond to abiotic stresses and regulate panicle development in rice [89]. The cis-element analysis revealed that there were many light response elements in the promoter regions of *SlPP2C* genes, such as sp1, G-box, ACE, and 3-AF1 binding sites. Among them, G-box appeared in 70 out of 92 *SlPP2Cs*, which may have an important influence on regulating tomato accumulation dry matter by light and action. Furthermore, a large number of stress- and hormone-related elements were also found, such as ABA-responsive elements (ABREs) that are responsive to ABA, drought, or salt signals [90], LTR is involved in low-temperature response and regulation [91], TCA-element and CGTCA-motif have good correlation with the expression levels after MeJA and SA treatment [92], respectively. Further elucidating the predicated and the possible functions of *SlPP2C* genes in transcriptional regulation, the result represented significant variation among *SlPP2C* genes and was mostly responsive to *R. solanacearum.* Therefore, *PP2C* genes play a very important role in abiotic stress or biotic stress. It is of great research significance.

## 5. Conclusions

In this study, the *PP2C* gene family in tomato was classified and general analysis of the 92 members in this family was carried out including the proteins’ physical and chemical properties, subcellular localization, evolutionary relationship, gene duplication, environmental pressure, gene structure, conserved domains, cis-acting elements, conserved motifs, and expression patterns. Most of them showed tissue and developmental stage-specific expression profiles, and some of them can be induced by biotic stress (*R. solanacearum*), indicating that the *SlPP2Cs* play an important role in plants. The results of this study laid a foundation for more in-depth genetic transformation and gene function analysis and were necessary to advance research on the stress resistance of tomato. In summary, the integration of our findings has provided a novel insight and unique features of *SlPP2C* genes, which is also important for accelerating the cloning of stress resistance genes in tomato. 

## Figures and Tables

**Figure 1 genes-13-00604-f001:**
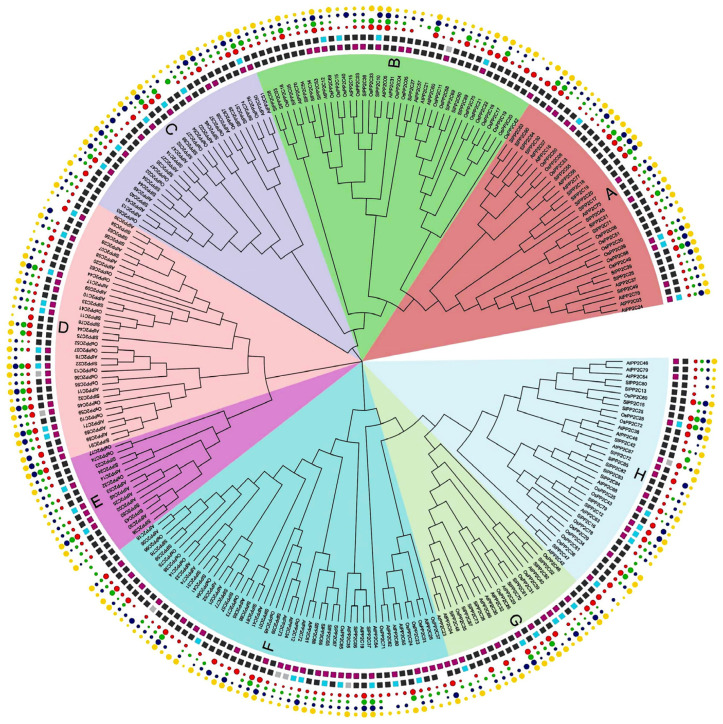
Phylogenetic analysis of the tomato *PP2C* family. It is based on protein sequence aligned by the ClustalW program. MEGA-X was used to construct a phylogenetic tree with the maximum likelihood method. Different colors indicate different subfamily members according to sequence similarity annotation analysis.

**Figure 2 genes-13-00604-f002:**
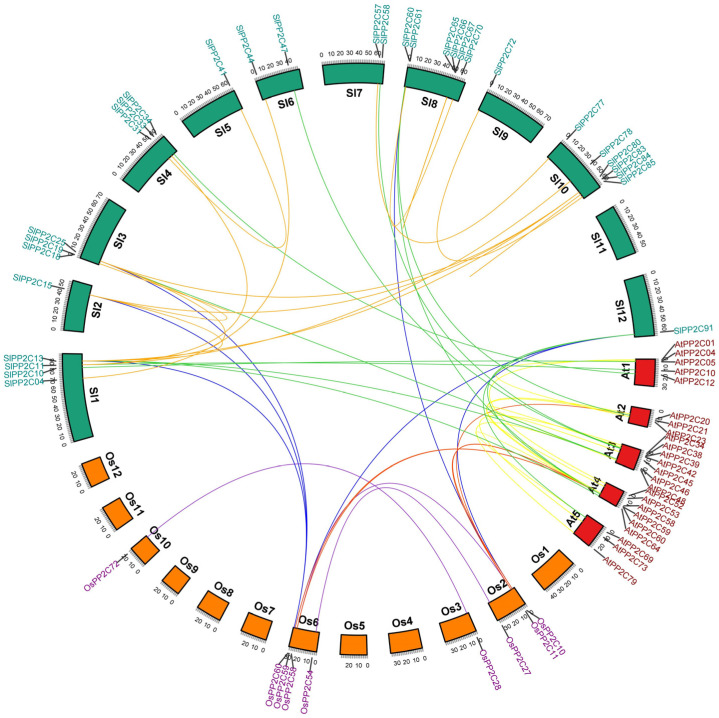
Duplication event analysis of *SlPP2C* genes and comparative synteny analysis among tomato, Arabidopsis, and rice, between tomato and Arabidopsis, and between tomato and rice. The red line represents the syntenic gene pairs between rice and Arabidopsis. The blue line indicates the syntenic gene pairs between rice and tomato. The green line represents the syntenic gene pairs between tomato and Arabidopsis. The golden line represents the syntenic gene pairs in tomato. The yellow line represents the syntenic gene pairs in Arabidopsis. The purple line represents the syntenic gene pairs in rice. Tomato chromosomes are depicted as green segments, and Arabidopsis and rice are shown in red and orange, respectively. The chromosome number and syntenic gene pairs are marked. The size of chromosomes was consistent with the actual pseudo-chromosome size. Positions are in Mb.

**Figure 3 genes-13-00604-f003:**
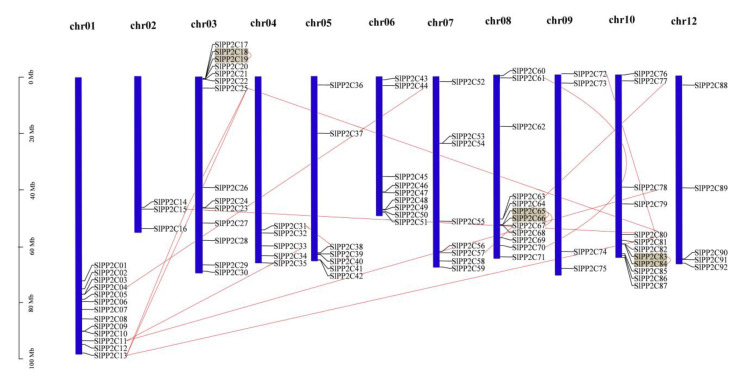
Chromosome distribution of tomato *PP2C* genes. Chromosome localization is based on the physical location (Mb) of 12 tomato chromosomes. Chromosome numbers are displayed at the top of each bar chart. Locations of tomato *PP2C* genes in chromosomes were obtained from the SGN (http://solgenomics.net (accessed on 5 March 2022)). Grey blocks were represented by the tandem duplicated genes, and the segmentally duplicated genes were linked by red lines. Scale bar on the left indicated the length (Mb) of tomato chromosomes.

**Figure 4 genes-13-00604-f004:**
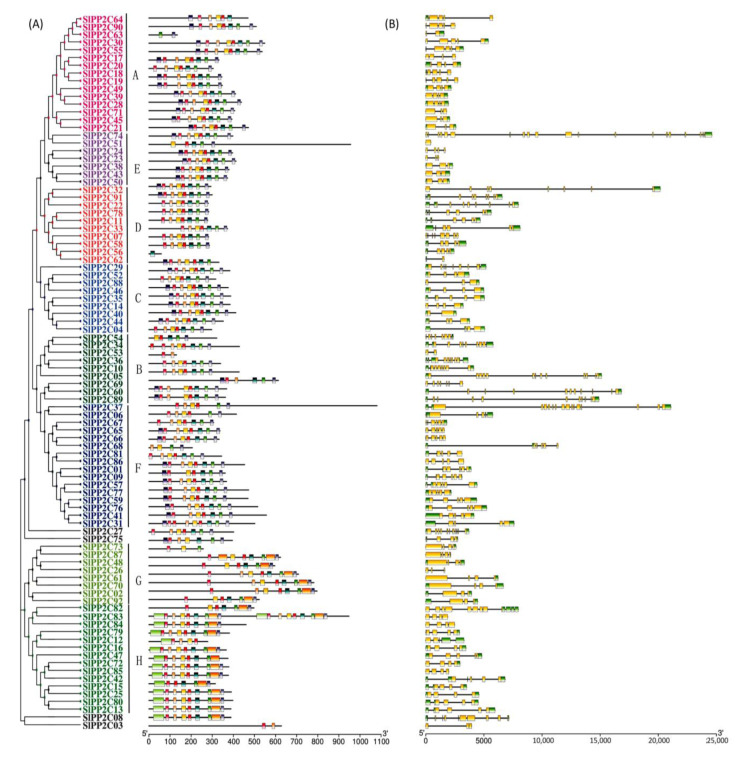
Phylogenetic relationships, conserved motifs of *SlPP2C* proteins, and structures of *SlPP2C* genes in tomato. (**A**) Arrangement of conserved motifs in *SlPP2C* proteins. Ten predicted motifs were represented by different colored boxes, and motif details referred to Table 3. A-H depicted that PP2C genes were divided into eight groups. Scale bar indicates amino acid length (**B**) Gene structure of *SlPP2C* members. The phylogenetic tree was constructed using MEGA-X software and the gene structures were visualized by TBtools. Boxes represented exons, and yellow boxes represented CDS and the upstream and downstream regions of *SlPP2C* genes were indicated by green boxes. For all genes, black lines represent introns. The sizes of genes can be estimated by the scale at the bottom.

**Figure 5 genes-13-00604-f005:**
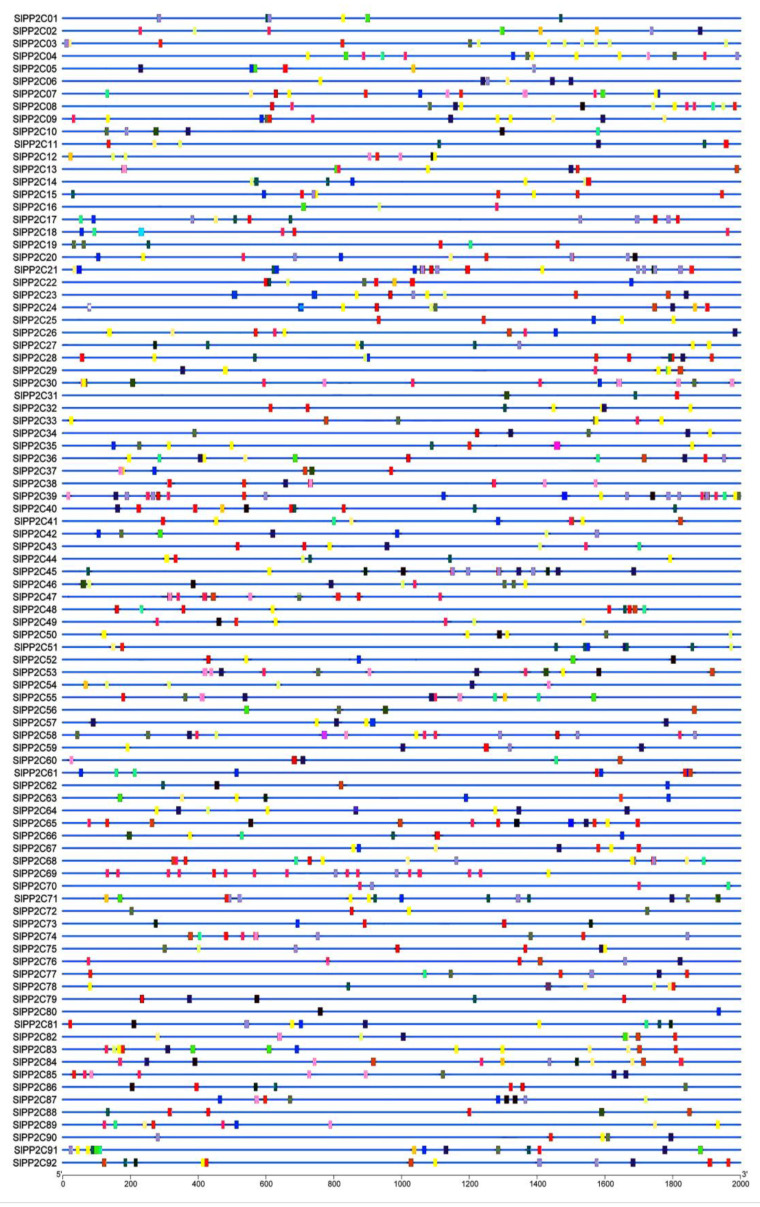
Identified cis-elements in the promoters of *SlPP2C* genes. The grey line represents the 2000 bp upstream of the *SlPPCs* transcription start site. Different colored wedges represent different cis-elements. The length and position of each *SlPP2C* gene are drawn to scale. Scale bar indicates DNA sequence length.

**Figure 6 genes-13-00604-f006:**
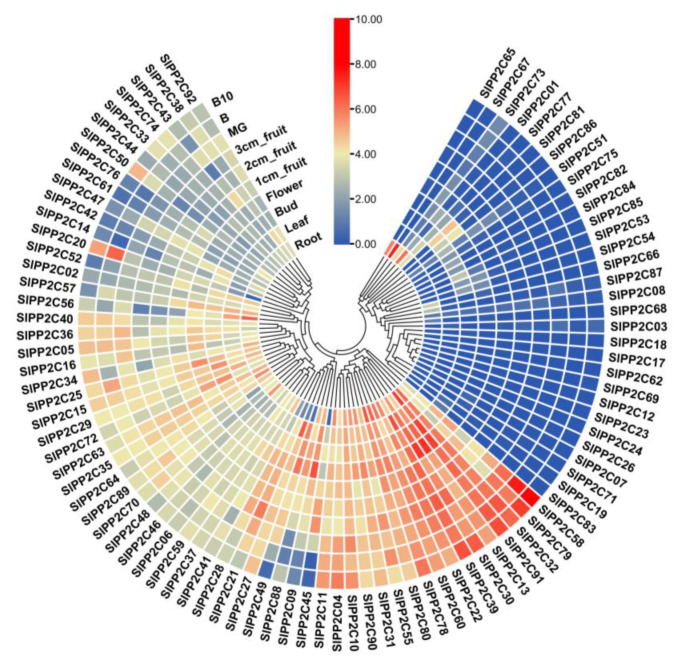
Heat map of the expression patterns of *SlPP2Cs* in 10 tissues/stages. Data of 10 tissues in the TFGD database were collected to reconstruct the expression pattern of *SlPP2C* genes. Heat map is presented in blue/yellow/red colors that indicate low/medium/high expression, respectively. The result was processed through cluster analysis.

**Figure 7 genes-13-00604-f007:**
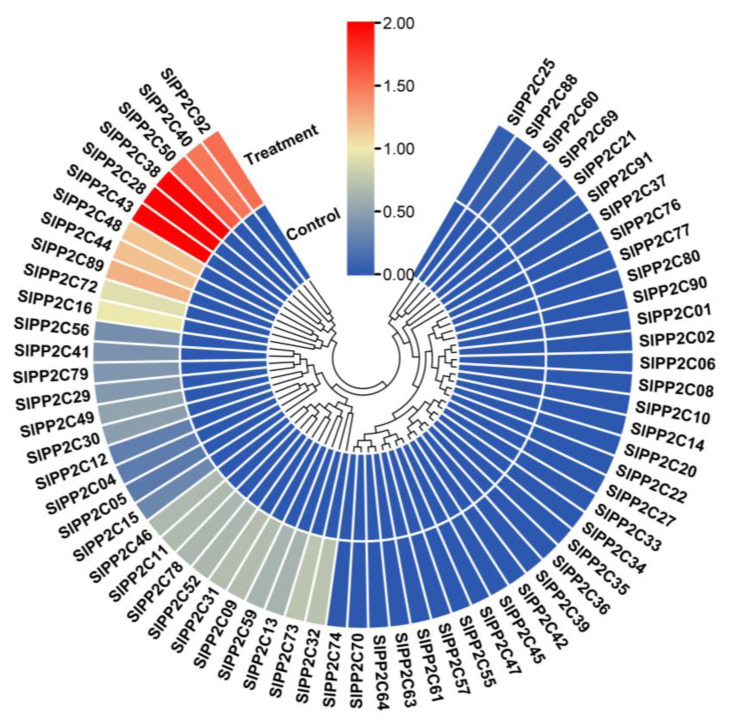
Expression profiles of *SlPP2C* genes in tomato roots infected by plant pathogen *Ralstonia solanacearum*. Blocks with colors represent decreased (blue) or increased (red) transcript levels relative to the control.

**Figure 8 genes-13-00604-f008:**
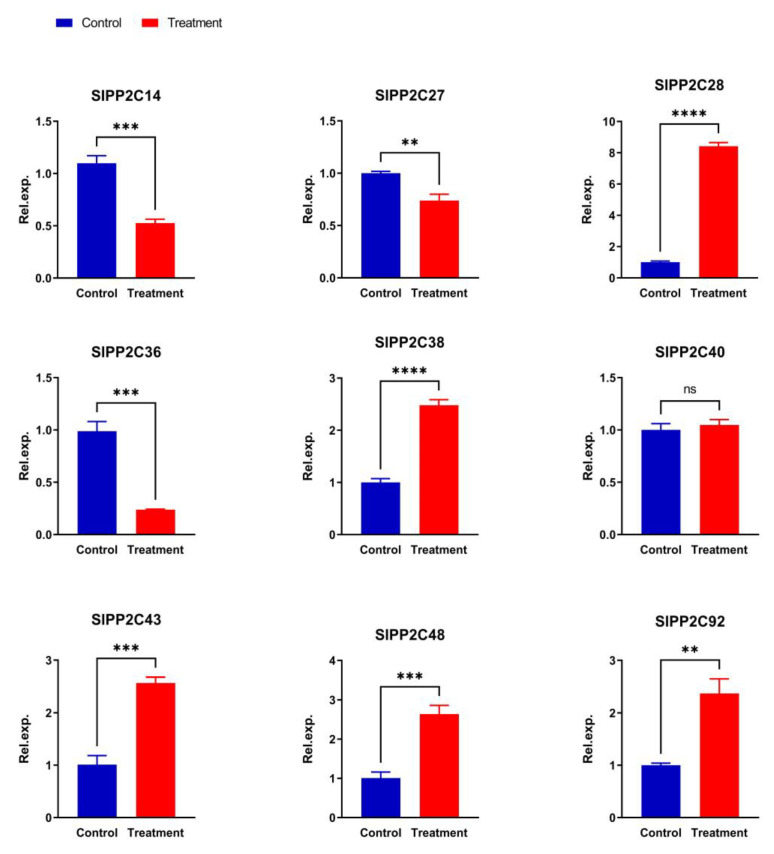
Expression of *SlPP2C* genes under plant pathogen *Ralstonia solanacearum* infection treatment. Control means untreated plants playing a controlling role. Treatment is *R. solanacearum* infection treatment plant. Expression of treated plants was compared with that in untreated plants after normalization of values with reference to the tomato *β-actin* gene and is presented as the relative expression level. All samples were collected from three biological replicates of each treatment at specified intervals. The error bars represent the SEM. ** *p* < 0.01, *** *p* < 0.001, **** *p* < 0.0001, Ns: not significant. The expression patterns of the selected *SlPP2C* genes were analyzed by qRT-PCR with gene-specific primers (Appendix A).

**Table 1 genes-13-00604-t001:** Information of the protein phosphatase 2C (*PP2C*) gene family in tomato.

Gene	Identifier	Chromosome	AA	PIs/MW	GRAVY	Instability	Subcellular
		Location				Index	Localization
*SlPP2C01*	Solyc01g065700.3.1	chr01: 72,289,658–72,295,088	361	6.47/39,992.45	−0.266	35.78	Cyto
*SlPP2C02*	Solyc01g066870.3.1	chr01: 750,92,877–75,095,866	522	5.09/57,125.35	−0.202	42.69	Cyto
*SlPP2C03*	Solyc01g067980.1.1	chr01: 77,027,259–77,032,781	202	6.19/20,650.27	−0.001	31.43	Extr. Nucl
*SlPP2C04*	Solyc01g068000.4.1	chr01: 77,050,710–77,057,722	297	4.89/32,913.15	−0.202	54.88	Nucl
*SlPP2C05*	Solyc01g079720.3.1	chr01: 78,822,308–78,843,121	613	6/68,185.28	−0.282	45.77	Cyto
*SlPP2C06*	Solyc01g080400.3.1	chr01: 79,645,915–79,653,551	414	4.81/45,150.79	−0.406	46.18	Nucl
*SlPP2C07*	Solyc01g087460.3.1	chr01: 82,424,646–82,428,543	372	6.17/41,583.98	−0.655	38.39	Nuc
*SlPP2C08*	Solyc01g094230.4.1	chr01: 85,781,740–85,791,508	627	11.39/64,450.04	−0.476	35.66	Nucl
*SlPP2C09*	Solyc01g100040.3.1	chr01: 90,110,898–90,115,136	367	6.22/40,919.44	−0.258	38.73	Nucl
*SlPP2C10*	Solyc01g100110.4.1	chr01: 90,196,815–90,202,470	427	6.91/46,135.74	−0.146	43.86	Cyto. Nucl
*SlPP2C11*	Solyc01g105280.3.1	chr01: 93,506,692–93,513,168	283	6.8/30,954.82	−0.426	36.53	Cyto. Nuc
*SlPP2C12*	Solyc01g107300.4.1	chr01: 94,888,760–94,892,503	278	7.79/31,335.87	−0.128	46.13	Cyto
*SlPP2C13*	Solyc01g111730.3.1	chr01: 97,915,696–97,923,710	388	8.23/42,809.56	−0.263	44.55	Chlo. Nucl
*SlPP2C14*	Solyc02g082490.4.1	chr02: 46,189,450–46193937	384	5.67/41,815.49	−0.313	50.46	Nucl
*SlPP2C15*	Solyc02g083420.3.1	chr02: 46,831,821–46,836,739	390	7.74/43,034.89	−0.226	48.53	Nucl
*SlPP2C16*	Solyc02g092750.3.1	chr02: 53,716,175–53,721,010	366	5.45/40,529.01	−0.229	39.32	Nucl
*SlPP2C17*	Solyc03g006930.3.1	chr03: 1,502,552–1,506,122	331	6.09/36,843.85	−0.25	59.95	Cyto
*SlPP2C18*	Solyc03g006940.3.1	chr03: 1,509,616–1,512,540	345	4.95/37,872.93	−0.185	57	Nucl
*SlPP2C19*	Solyc03g006950.3.1	chr03: 1,514,383–1,518,178	347	6.02/38,049.53	−0.147	53.69	Nucl
*SlPP2C20*	Solyc03g006960.4.1	chr03: 1,519,421–1,523,583	304	5.7/33,397.94	−0.255	56.99	Cyto
*SlPP2C21*	Solyc03g007230.4.1	chr03: 1,791,778–17,95,346	397	5.72/43,979.53	−0.424	50.8	Nucl
*SlPP2C22*	Solyc03g007270.3.1	chr03: 1,828,441–1,838,542	299	4.96/32,489.18	−0.432	43.54	Nucl
*SlPP2C23*	Solyc03g013100.1.1	chr03: 47,516,649–47,518,201	114	6.51/12,939.8	−0.337	35.03	Mito
*SlPP2C24*	Solyc03g013140.1.1	chr03: 47,332,347–47,334,696	163	5.81/18,565.16	−0.428	20.18	Cyto
*SlPP2C25*	Solyc03g033340.3.1	chr03: 4,912,004–4,918,389	397	7.65/43,970.18	−0.217	47.54	Chlo. Nucl
*SlPP2C26*	Solyc03g065190.1.1	chr03: 40,216,058–40,218,346	123	9.55/14,453	−0.293	32.52	Cyto. Extr. Nucl
*SlPP2C27*	Solyc03g082960.2.1	chr03: 52,814,842–52,820,023	395	6.29/43,427.5	−0.203	44.71	Nucl
*SlPP2C28*	Solyc03g096670.3.1	chr03: 58,989,562–58,992,297	406	6/44,683.73	−0.397	52.59	Cyto. Nucl
*SlPP2C29*	Solyc03g118890.4.1	chr03: 67,672,668–67,679,525	329	6.22/36,093.73	−0.248	42.26	Cyto
*SlPP2C30*	Solyc03g121880.4.1	chr03: 69,911,090–69,918,573	548	5.86/59,716.58	−0.095	38.86	Cyto
*SlPP2C31*	Solyc04g056560.4.1	chr04: 54,528,741–54,538,177	401	4.87/43,785.08	−0.266	32.37	Nucl
*SlPP2C32*	Solyc04g064500.4.1	chr04: 55,645,130–55,673,295	372	4.95/40,544.4	−0.481	45.68	Nucl
*SlPP2C33*	Solyc04g074190.3.1	chr04: 60,189,483–60,198,295	278	6.32/30,483.44	−0.345	49.68	Cyto. Nucl
*SlPP2C34*	Solyc04g079120.3.1	chr04: 63,704,094–63,711,682	428	5.28/46,166.21	−0.296	47.94	Nucl
*SlPP2C35*	Solyc04g082600.3.1	chr04: 66,227,666–66,234,622	387	4.85/42,607.25	−0.265	47.37	Chlo. Nucl
*SlPP2C36*	Solyc05g009070.4.1	chr05: 3,216,982–3,221,083	339	8.01/37,083.37	−0.231	35.43	Nucl
*SlPP2C37*	Solyc05g018300.3.1	chr05: 20,358,675–20,387,085	1080	4.81/119,917.55	−0.203	42.53	Nucl
*SlPP2C38*	Solyc05g052520.3.1	chr05: 62,745,936–62,749,176	396	5.56/42,845.56	−0.233	53.55	Nucl
*SlPP2C39*	Solyc05g052980.4.1	chr05: 63,149,350–63,151,994	437	5.69/48,147.65	−0.281	52.66	Nucl
*SlPP2C40*	Solyc05g053290.3.1	chr05: 63,421,852–63,425,476	411	4.73/44,751.23	−0.109	43.03	Nucl
*SlPP2C41*	Solyc05g055790.4.1	chr05: 65,291,032–65,295,556	499	5.23/55,284.13	−0.375	52.06	Nucl
*SlPP2C42*	Solyc05g055980.4.1	chr05: 65,383,352–65,389,926	314	6.91/35,158.87	−0.366	43.45	Chlo. Nucl
*SlPP2C43*	Solyc06g007190.4.1	chr06: 1,265,243–1,268,111	412	7.97/4,4889.01	−0.182	44.84	Nucl
*SlPP2C44*	Solyc06g009390.3.1	chr06: 3,311,448–3,316,710	352	4.94/38,881.06	−0.11	56.6	Nucl
*SlPP2C45*	Solyc06g051940.4.1	chr06: 35,603,081–35,605,882	471	5.8/51,885.49	−0.36	43.54	Nucl
*SlPP2C46*	Solyc06g065610.3.1	chr06: 40,985,604–40,992,533	375	5.59/40,805.91	−0.342	45.11	Nucl
*SlPP2C47*	Solyc06g065920.4.1	chr06: 41,312,022–41,318,633	374	8.51/41,712.42	−0.298	46.39	Nucl
*SlPP2C48*	Solyc06g076100.3.1	chr06: 47,243,874–47,248,381	708	5.41/79,199.49	−0.537	38.2	Nucl
*SlPP2C49*	Solyc06g076400.3.1	chr06: 47,471,880–47,474,601	410	5.12/44,859.67	−0.346	46.42	Nucl
*SlPP2C50*	Solyc06g082080.3.1	chr06: 48,014,366–48,017,179	379	8.18/40,564.02	−0.241	52.55	Nucl
*SlPP2C51*	Solyc06g082700.1.1	chr06: 48,429,677–48,430,277	142	6.09/15,813.08	−0.125	26.13	Cell membrane. Chlo. Cyto
*SlPP2C52*	Solyc07g007220.3.1	chr07: 1,956,469–1,961,556	383	5.07/42,272.7	−0.264	47.32	Nucl
*SlPP2C53*	Solyc07g024010.2.1	chr07: 23,836,770–23,838,030	136	6.07/15,409.59	−0.258	24.63	Chlo
*SlPP2C54*	Solyc07g024020.2.1	chr07: 23,848,754–23,851,910	321	5.09/34,819.28	−0.321	40.61	Nucl
*SlPP2C55*	Solyc07g040990.4.1	chr07: 51,484,565–51,489,081	536	5.38/59,196.43	−0.267	47.81	Nucl
*SlPP2C56*	Solyc07g053760.4.1	chr07: 62,192,070–62,195,463	286	6.67/31,303.63	−0.331	37.81	Nucl
*SlPP2C57*	Solyc07g054300.3.1	chr07: 62,654,289–62,659,599	478	5.38/53,166.78	−0.45	44.34	Nucl
*SlPP2C58*	Solyc07g062970.3.1	chr07: 65,595,765–65,600,127	282	5.67/30,943.93	−0.327	28.7	Nucl
*SlPP2C59*	Solyc07g066260.3.1	chr07: 67,742,920–67,748,833	515	5.75/56,463.59	−0.371	38.51	Chlo. Nucl
*SlPP2C60*	Solyc08g006060.3.1	chr08: 803,615–827,195	368	4.99/40,229.15	−0.398	38.43	Nucl
*SlPP2C61*	Solyc08g007000.3.1	chr08: 1,574,889–1,583,560	781	5.23/86,116.34	−0.491	47.03	Chlo
*SlPP2C62*	Solyc08g044610.3.1	chr08: 18,794,957–18,799,764	59	6.71/6738.81	0.012	42.27	Chlo. Nucl.
*SlPP2C63*	Solyc08g062640.3.1	chr08: 51,623,577–51,625,677	134	7.61/15,444.01	−0.244	40.43	Nucl.
*SlPP2C64*	Solyc08g062650.2.1	chr08: 51,624,426–51,632,488	469	4.89/51,638.63	−0.057	38.97	Nucl
*SlPP2C65*	Solyc08g065500.2.1	chr08: 53,632,222–53,634,431	336	5.06/36,816.81	−0.137	40.85	Nucl
*SlPP2C66*	Solyc08g065540.3.1	chr08: 53,677,271–53,679,628	332	5.59/36,782.71	−0.185	39.99	Nucl
*SlPP2C67*	Solyc08g065670.4.1	chr08: 53,898,474–53,901,007	306	5.22/33,454.16	−0.075	40.12	Nucl
*SlPP2C68*	Solyc08g065680.3.1	chr08: 53,924,816–53,927,725	205	5.24/22,537.35	−0.166	44.21	Nucl
*SlPP2C69*	Solyc08g074230.1.1	chr08: 58,350,631–58,355,073	271	4.76/30,183.9	−0.518	37.99	Nucl
*SlPP2C70*	Solyc08g077150.3.1	chr08: 61,046,865–61,055,706	796	5.19/87,391.85	−0.456	50.57	Chlo
*SlPP2C71*	Solyc08g082260.2.1	chr08: 65,099,205–65,101,710	393	6.17/44,173.95	−0.55	45.52	Nucl
*SlPP2C72*	Solyc09g007080.3.1	chr09: 727,617–731,731	378	9.06/41,932.11	−0.266	41.45	Chlo. Cyto. Mito. Nucl
*SlPP2C73*	Solyc09g010780.3.1	chr09: 4,070,320–4,073,958	623	5.4/69,387.69	−0.394	29.6	Chlo
*SlPP2C74*	Solyc09g065650.3.1	chr09: 63,821,833–63,855,480	955	5.43/109,110.03	−0.408	38.77	Nucl
*SlPP2C75*	Solyc09g090280.3.1	chr09: 69,809,431–69,813,531	257	9.78/28,883.02	−0.807	42.35	Chlo. Nucl
*SlPP2C76*	Solyc10g005640.4.1	chr10: 513,071–520,386	556	5.56/61,356.83	−0.502	41.89	Nucl
*SlPP2C77*	Solyc10g008490.3.1	chr10: 2,617,430–2,620,523	469	5.02/51,815.01	−0.455	46.09	Nucl
*SlPP2C78*	Solyc10g047290.2.1	chr10: 40,367,471–40,374,422	281	8.32/30,574.56	−0.431	40.11	Nucl
*SlPP2C79*	Solyc10g049630.2.1	chr10: 46,299,689–46,303,378	381	9.06/42,379.44	−0.334	48.38	Nucl
*SlPP2C80*	Solyc10g055650.2.1	chr10: 57,152,293–57,156,799	388	8.67/43,047.9	−0.271	42.35	Nucl. Pero
*SlPP2C81*	Solyc10g076320.3.1	chr10: 59,239,617–59,243,965	344	6.05/38,842.14	−0.424	40.32	Chlo. Nucl
*SlPP2C82*	Solyc10g078800.3.1	chr10: 60,493,362–60,504,498	947	5.26/105,611.28	−0.386	45.49	Nucl
*SlPP2C83*	Solyc10g078810.1.1	chr10: 60,504,641–60,507,279	438	4.93/48,492.24	−0.355	39.39	Nucl
*SlPP2C84*	Solyc10g078820.2.1	chr10: 60,508,870–60,512,342	460	4.99/51,210.32	−0.422	46.42	Nucl
*SlPP2C85*	Solyc10g084410.2.1	chr10: 63,960,722–63,963,486	376	9.42/41,906.86	−0.346	47.56	Nucl
*SlPP2C86*	Solyc10g085370.3.1	chr10: 64,570,827–64,575,317	453	5.86/51,847.37	−0.654	36.8	Nucl
*SlPP2C87*	Solyc10g086490.2.1	chr10: 65,295,550–65,298,502	596	5.67/66,435.52	−0.392	39	Cell membrane. Chlo
*SlPP2C88*	Solyc12g010450.3.1	chr12: 3,470,481–3,476,558	316	5.75/34,814.63	−0.272	43.32	Cyto
*SlPP2C89*	Solyc12g042570.2.1	chr12: 39,939,943–39,959,306	362	4.92/39,315.91	−0.512	39.46	Nucl
*SlPP2C90*	Solyc12g096020.3.1	chr12: 65,098,300–65,101,758	508	4.62/55,681.53	−0.235	45.3	Nucl
*SlPP2C91*	Solyc12g096520.3.1	chr12: 65,400,452–65,408,533	293	5.04/31,552.91	−0.349	34.47	Cyto
*SlPP2C92*	Solyc12g099600.2.1	chr12: 66,688,468–66,694,648	497	5.35/53,856.74	−0.189	47.82	Nucl

AA: number of amino acids; pIs: theoretical isoelectric point; MW: molecular weight (kDa). GRAVY: grand average of hydropathicity (GRAVY < 0, hydrophilic protein/GRAVY > 0, hydrophobic protein), Instability Index (<40, the protein is stable/>40, the protein is unstable). Cyto: cytoplasm; Extr: extracellular; Nucl: nucleus; Chlo: chloroplast; Mito: mitochondria; Pero: peroxisome.

**Table 2 genes-13-00604-t002:** The distribution of *PP2C* genes in Arabidopsis, rice, and tomato.

Subgroup of *PP2C* Genes	Numbers of *AtPP2Cs*	Numbers of *OsPP2Cs*	Numbers of *SlPP2Cs*
A	9	9	15
B	8	18	12
C	9	8	11
D	13	11	11
E	6	3	5
F	19	14	16
G	7	5	8
H	9	10	14

*AtPP2Cs: Arabidopsis thaliana PP2Cs; OsPP2Cs: Oryza sativa PP2Cs; SlPP2Cs: Solanum lycopersicum PP2Cs.*

**Table 3 genes-13-00604-t003:** *SlPP2C* syntenic gene pairs present in tomato genome.

Gene ID	Ka	Ks	Ka/Ks	Divergence Time (Myr)
*SlPP2C04/44*	0.09	0.69	0.12	23
*SlPP2C11/78*	0.1	0.54	0.19	18
*SlPP2C13/80*	0.08	0.63	0.12	21
*SlPP2C15/25*	0.07	0.6	0.11	20
*SlPP2C15/12*	0.12	2.04	0.06	68
*SlPP2C15/80*	0.14	2.2	0.06	73.33
*SlPP2C18/19*	0.08	0.09	0.93	3
*SlPP2C25/13*	0.14	2.01	0.07	67
*SlPP2C33/11*	0.19	1.68	0.11	56
*SlPP2C41/31*	0.28	0.85	0.33	28.33
*SlPP2C57/57*	0.09	0.75	0.12	25
*SlPP2C61/70*	0.09	0.72	0.12	24
*SlPP2C66/65*	0.08	0.37	0.22	12.33
*SlPP2C67/65*	0.02	0.07	0.26	2.33
*SlPP2C80/25*	0.15	1.93	0.08	64.33
*SlPP2C84/83*	0.08	0.12	0.64	4
*SlPP2C85/72*	0.12	0.82	0.15	27.33

**Table 4 genes-13-00604-t004:** Conserved motifs in the amino acid sequences of *SlPP2C* proteins.

Motif	Width	Multilevel Consensus Sequence
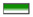 1	15	*FLILASDGLWDVLSN*
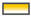 2	21	*VIQGETLYVANVGDSRAVLCR*
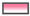 3	15	*TFFGVYDGHGGPGAA*
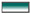 4	20	*VWRVKGGLAVSRAIGDKYLK*
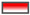 5	15	*AIQLSVDHKPNREDE*
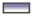 6	15	*RGSHDBISVIVVFLD*
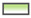 7	50	*HEGGDLGGRQDGLLWYKDLGQHANGEFSMAVVQANNLLEDQSQVESGPLS*
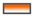 8	41	*AVDIVHSYPRGGIARRLVKAALQEAAKKREMRYSDLKKIDR*
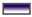 9	20	*SQQGRRGEMEDAHIVWPBFC*
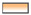 10	14	*KKALRKAFLKTDEE*

**Table 5 genes-13-00604-t005:** Functionally annotated cis-elements identified in the promoters of 92 *SlPP2Cs*.

Cis-Element	Number of Genes	Functions of Cis-Elements
	circadian	19	circadian control
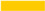	GT1-motif	57	light responsiveness
	G-box	70	light responsiveness
	MRE	36	light responsiveness
	ACE	21	light responsiveness
	3-AF1	10	light responsive
	Sp1	9	light responsive
	4cl-CMA2b	1	light responsive
	AAAC-motif	3	light responsive
	CGTCA-motif	60	MeJA-responsiveness
	TGACG-motif	60	MeJA-responsiveness
	ABRE	70	abscisic acid responsiveness
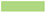	TGA-	26	auxin-responsive
	AuxRR-core	11	auxin responsiveness
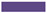	TCA-	47	salicylic acid responsiveness
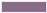	SARE	1	salicylic acid responsiveness
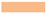	GARE-motif	21	gibberellin-responsive
	P-box	31	gibberellin-responsive
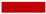	TATC-box	16	gibberellin-responsiveness
	LTR	24	low-temperature responsiveness
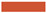	MBS	38	drought-inducibility
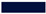	MBSI	8	flavonoid biosynthetic genes regulation
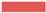	MSA-like	4	cell cycle regulation
	O2-site	25	zein metabolism regulation
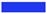	Box III	4	protein binding site

## Data Availability

Not applicable.

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
