# Peer review of "Genome-Wide Analysis of the Protein Phosphatase 2C Genes in Tomato"

_genes, 2022, doi:10.3390/genes13040604_

Round 1
Reviewer 1 Report
Comments are directly given in the reviewed article PDF.

Author Response
Dear reviewer:
We gratefully appreciate you for reading our paper carefully. According with your advice, we amended the relevant part in manuscript. Please see the attachment.
Heart-felt thanks again for your careful check to our manuscript!
Sincerely,
Best wishes!

Reviewer 2 Report
In my opinion Authors present important research results of protein phosphatase 2C gene family in tomato. Authors studied their phylogeny, expression, gene organization and response to pathogen (Ralstonia solanacearum) infection. Study is well planned and performed. Conclusions are supported by research results.
Line 26, 32- name Ralstonia solanacearum should be in italics
Line 32 doubled sign : (remove one of them)
Line 130- double spaced after 28⁰C
Sentence 131-133 should be rewritten
138,146 and other, space before ⁰C. remove it
Line 140- is m2, should be m2
μl, ml- should be corrected to μL, mL in the entire text
line 151- amount of RNA taken for experiment
Line 159- double spaced
Line 160 - Citation of Livak and Schmittgen method used for relative gene expression calculation should be included.
Line 203- Authors write of a 2kb long regions, localized upstream to ATG, while in line 24 it is 3 kb, correct it. In line 365 it is 3 kb.
Section 2.7
Authors performed searches for cis-active elements within the promoters- 2kb upstream from the translation initiation site. However, Authors should be aware, that most of the in vivo functional cis-active elements (about 80%) is concentrated within proximal promoters. Authors could add one or two sentences to show this fact in section 2.7 or Discussion. Sample reference is below:
Yu ChP, Lin JJ, Li WH (2016) Positional distribution of transcription factor binding sites in Arabidopsis thaliana. Sci Rep 6:25164.
Author Response
Dear reviewer:
I am very grateful to your comments for our manuscript. According with your advice, we amended the relevant part in manuscript. Please see the attachment.
Heart-felt thanks again for your careful check to our manuscript!
Sincerely,
Best wishes!

Reviewer 3 Report
This article shows some interesting results on genome-wide analysis of the protein phosphatase 2C genes in tomato. Authors have analyzed many genes in protein phosphatase 2C gene family. However, some Figures are not readable well. So Tables need to be made to provide the detailed information derived from the Figure results.
Line 27: RNA-seq and qRT-PCR date --- data
Lines 127~226: in “Materials and Methods” should be re-ordered according to the order of “Results”.
In section “3.2. Phylogenetic and comparative synteny analysis”, Figure 1 is not readable. Figure 1 should be provided in much larger image. As it is, Figure 1 gives readers no information at all. So authors should provide the grouping results as a Supplementary Table in MS Word.
Figure 2. Duplication event analysis of SlPP2C genes and comparative synteny analysis among tomato, Arabidopsis and rice, between tomato and Arabidopsis, and between tomato and rice. Also Figure 2 has the same situation as Figure 1. Figure 2 should be provided in much larger image. As it is, Figure 2 gives readers no information at all. So authors should provide the grouping results as a Supplementary Table in MS Word.
Figure 3. Chromosome distribution of tomato PP2C genes. The map is not readable. So it must be provided as a larger one.
Table 3 and Figure 3, The colors that are similar are not distinguishable, they need to be clearly detectable.
Also, the results in Figure 3 is not visible, it must be much larger than the current one.
I would ask authors to provide bigger image of Figure 5.
There is about 18% plagiarism detected, so some parts need to be rewritten.
Author Response
Dear reviewer:
I am very grateful to your comments for our manuscript. According with your advice, we amended the relevant part in manuscript. Please see the attachment.
Heart-felt thanks again for your careful check to our manuscript!
Sincerely,
Best wishes!

This manuscript is a resubmission of an earlier submission. The following is a list of the peer review reports and author responses from that submission.